# Non-Negative Matrix Tri-Factorization for Representation Learning in Multi-Omics Datasets with Applications to Drug Repurposing and Selection

**DOI:** 10.3390/ijms25179576

**Published:** 2024-09-04

**Authors:** Letizia Messa, Carolina Testa, Stephana Carelli, Federica Rey, Emanuela Jacchetti, Cristina Cereda, Manuela Teresa Raimondi, Stefano Ceri, Pietro Pinoli

**Affiliations:** 1Department of Electronics, Information and Bioengineering (DEIB), Politecnico di Milano, 20133 Milan, Italy; 2Center of Functional Genomics and Rare Diseases, Buzzi Children’s Hospital, 20154 Milan, Italy; 3Pediatric Clinical Research Center “Fondazione Romeo ed Enrica Invernizzi”, Department of Biomedical and Clinical Sciences, Università degli Studi di Milano, 20157 Milan, Italy; 4Department of Chemistry, Materials and Chemical Engineering “Giulio Natta”, Politecnico di Milano, 20133 Milan, Italy

**Keywords:** representation learning, machine learning, data integration, drug repurposing, drug selection, personalized medicine

## Abstract

The vast corpus of heterogeneous biomedical data stored in databases, ontologies, and terminologies presents a unique opportunity for drug design. Integrating and fusing these sources is essential to develop data representations that can be analyzed using artificial intelligence methods to generate novel drug candidates or hypotheses. Here, we propose Non-Negative Matrix Tri-Factorization as an invaluable tool for integrating and fusing data, as well as for representation learning. Additionally, we demonstrate how representations learned by Non-Negative Matrix Tri-Factorization can effectively be utilized by traditional artificial intelligence methods. While this approach is domain-agnostic and applicable to any field with vast amounts of structured and semi-structured data, we apply it specifically to computational pharmacology and drug repurposing. This field is poised to benefit significantly from artificial intelligence, particularly in personalized medicine. We conducted extensive experiments to evaluate the performance of the proposed method, yielding exciting results, particularly compared to traditional methods. Novel drug–target predictions have also been validated in the literature, further confirming their validity. Additionally, we tested our method to predict drug synergism, where constructing a classical matrix dataset is challenging. The method demonstrated great flexibility, suggesting its applicability to a wide range of tasks in drug design and discovery.

## 1. Introduction

In recent years, we have been witnessing a steady increase of available biomedical omics data for investigation. Given the abundance and heterogeneity of such data, computational tools capable of integrating multiple sources of information to generate concise and semantically meaningful representations of entities (e.g., genes or drugs) are invaluable. These representations can be exploited to leverage artificial intelligence (AI) methods which are posed to play an increasingly crucial role in biomedical fields. Computational pharmacology, particularly drug design and repurposing, is one of those fields that greatly benefits from data integration and AI. Drug repurposing, specifically, aims at identifying new therapeutic applications for existing approved drugs, offering a novel cost-effective alternative to traditional drug development. Indeed, conventional approaches to develop novel drugs face many limitations. First, they require substantial financial burdens, with an average cost of bringing a new drug to market that hovers around USD 1 billion [1]. Moreover, these approaches adhere to a defined trajectory, encompassing four phases: drug discovery, pre-clinical evaluations, clinical trials, and post-marketing safety monitoring. These stages are complex and rigorous, with strict examination of safety, effectiveness, and quality requiring typically from 12 to 17 years [2]. In addition, the majority of newly identified molecules fail in the early stage of clinical trials, with only 14% of initially identified potential drugs receiving approval from the authorities [3]. Thus, drug repurposing is rapidly emerging as a innovative and cost-effective alternative. It entails the assignment of already approved drugs, originally developed for one medical condition, for the treatment of different conditions. The benefits of this approach stem from a notable reduction in both time and financial investment (down to USD 8 million and 3 to 12 years), along with an higher success rate (up to 30–75%) [4].

Drug repurposing can be achieved both experimentally, via laboratory-based investigations, and computationally, through data-driven means [5]. Experimental-driven drug repurposing often arises serendipitously and is typically not driven by a defined hypothesis on the target [6]. It might result from an experimental drug screening or by identifying target similarities among different diseases. In contrast, in silico drug repurposing is definitely driven by technological advancements, such as the generation of high-throughput data from diverse sources, leading to the accessibility of increasingly comprehensive datasets, and the advancements in AI methods [5]. This approach uniquely converts system biology data about phenotypes and targets into a prediction of druggable target/s and, ideally, provides FDA-approved compounds with potential modulatory/inhibitory functions [7]. Computational drug repurposing strategies can be classified in several ways, with the main distinction between strategies being focused on discovering new applications for existing drugs (drug-centric) and those aimed at identifying effective drugs for a particular disease (disease-centric) [8]. Both approaches share a common strategy of retrieving knowledge and assessing similarities between drugs and/or diseases [9].

Among computational methods, machine learning (ML), network models, and text mining are widely used.

ML techniques, such as logistic regression (LR), support vector machine (SVM), random forest (RF), neural networks (NNs), and deep learning (DL), have been applied extensively [10,11]. For instance, PREDICT [12] and SPACE [13] employ LR based on similarity for drug prediction. The first integrates drug–drug and disease–disease similarities to predict drugs with similar properties for comparable diseases, while the second forecasts the therapeutic chemical class of a drug by incorporating data from multiple sources. The SVM approach of Napolitano et al. [14] amalgamates drug–drug similarity datasets based on gene expression signatures, chemical structures, and molecular targets, utilizing unified drug similarity matrices as a kernel for multi-class SVM training. Wang et al. [15] employed a kernel function to correlate drugs with diseases, integrating molecular structure, activity, and phenotype data for SVM classifier training to predict novel drug–disease interactions. Menden et al. [16] fused cell line genomics and drug chemical structures, constructing a feed-forward perceptron neural network model and an RF regression model to predict cancer responses to drug treatments. The exploration of deep neural networks for drug repurposing in studies by Aliper et al. [17], Altae-Tran et al. [18], Hu et al. [19], and Segler et al. [20] showcases the potential of DL in the identification of novel therapeutic indications.

Network models represent entities such as drugs, genes, and proteins as nodes, with edges denoting connections. These models effectively capture interactions and relationships in biological and biomedical objects. Yamanishi et al. [21] proposed a bipartite graph supervised learning model using protein–protein interaction data, drug chemical structures, and drug–target interaction networks. Kinnings et al. [22] constructed a drug–drug network based on chemical structure and drug–target interaction similarity, revealing communities of drugs and uncovering therapeutic potentials. Hu and Agarwal [23] exploited microarray gene expression profiles to build a disease–drug network, identifying drug repositioning opportunities and potential side effects. Li and Lu [24] introduced a bipartite graph model inferring drug–target indications by considering drug pairwise similarity. Graphs of related and associated information can be effectively analyzed using Graph Neural Networks (GNNs). In 2022, Doshi and Sundeep [25] proposed a GNN-based architecture capable of analyzing graphs of drugs, indications, and related data to infer novel associations, with a specific application to COVID-19. A similar approach was also proposed in 2023 [26] and applied to a larger set of phenotypes.

Text mining methods harness the vast repository of information in biomedical and pharmaceutical literature to identify potential indications for existing drugs [27,28]. Grounded in biological ontology, these schemes allow for the comparison and analysis of biological information from diverse sources. Numerous text mining approaches for drug repurposing incorporate semantic inference technologies [29,30,31].

Another area that greatly benefits from AI is personalized medicine, which aims to provide more precise treatment by tailoring healthcare to the unique attributes of each individual. Personalized medicine involves integrating patient information, including genomic data, and analyze them with AI methods [32]. Multipartite graphs and Non-Negative Matrix Tri-Factorization (NMTF) are effective for this purpose.

Our research focuses on advancing in silico drug repurposing methodologies using advanced AI techniques. Specifically, we contribute to the field by developing and refining NMTF approaches to learn semantic representations of entities within complex multipartite graphs. These representations enable accurate predictions of novel drug–disease interactions, drug targets, synergistic relationships between drugs, and other critical biomedical insights [33,34,35,36,37,38]. Recently, we introduced a novel approach that exploits NMTF to only learn semantic representations of graph entities rather than directly infer novel associations [38], surpassing the performance of traditional NMTF. Similarly to NMTF, this method analyzes a multipartite graph (i.e., a graph in which the nodes are partitioned into disjoint sets and edges only connect nodes from different sets) to predict novel links between elements. In contrast, it utilizes NMTF solely for learning dense representations of the nodes. These representations are then employed in a “two-tower architecture”, where they are concatenated and fed into an AI algorithm. In our current study, we build upon our preliminary study, focusing on several key aspects for further development. Firstly, we extended the method to cope with hypergraphs (i.e., graphs where edges can connect any number of nodes). This required a transition to a “multi-tower architecture”. Furthermore, we ran an extensive test campaign to validate the method, simulating different scenarios: prediction of unseen drug–target interactions and synergistic relationships between pairs of anticancer drugs. Our approach demonstrated robust performance in predicting unseen drug–target interactions and synergistic relationships between anticancer drugs, particularly in the contexts of Parkinson’s disease (PD) and metastatic cancer. Additionally, a notable challenge arose with NMTF when attempting to predict the entire gene profile for genes lacking prior connections to any known drug, and, conversely, the complete drug profile for drugs lacking prior connections to any known gene. To address this challenge, we introduced the leave-one-out (LOO) validation technique, applied to both drugs and genes. This technique simulates scenarios in which a new drug or gene is introduced, and its interaction with existing entities is unknown. This validation confirmed our method’s advantages over conventional NMTF in both performance and applicability.

In summary, our contributions underscore the transformative potential of AI-driven approaches in personalized medicine and computational pharmacology, where the integration of heterogeneous data sources and advanced modeling techniques opens new avenues for more precise and efficient therapeutic strategies. Indeed, our work presents a computational method that can be applied to a wide variety of scenarios. Such a method leverages NMTF to integrate semantic and/or experimental information about various biomedical entities (e.g., drugs, genes, and cell lines) in order to learn a representation for these entities, rather than directly computing their interactions and associations. In particular, by utilizing gene expression profiles of patients suffering from specific conditions, our method—once validated from an experimental and biological standpoint—promises to significantly advance personalized medical treatments. This approach allows us to tailor therapies based on the unique genetic makeup of each patient, thus moving towards a more precise and individualized form of medicine. By integrating gene expression data with our AI-driven methodology, it can identify potential drug targets and therapeutic strategies that are specifically suited to the genetic profile of individual patients. The integration of this method into clinical practice could revolutionize the way we approach disease treatment and management, offering a powerful tool for the development of personalized therapies.

## 2. Results

### 2.1. Drug–Target Prediction

Here, we present the results of our methodology in the context of drug–target prediction. As use cases, our approach was applied to two distinct datasets, each relevant to significant medical conditions: metastatic cancers and PD. These conditions were selected due to their complex biological mechanisms and the critical need for targeted therapeutic strategies. The following subsections detail the selection criteria for genes and drugs included in the study, along with predictive performance, and potential implications of our findings within these two contexts. We discuss the robustness and accuracy of our predictions, along with comparisons to existing methodologies, thereby highlighting the advantages and limitations of our approach.

#### 2.1.1. Metastases Dataset

Conventional cancer treatments are primarily focused on eliminating or reducing the primary tumor mass through methods such as surgery, radiation, and chemotherapy. However, these treatments often fall short when it comes to address metastases, which are responsible for the majority of cancer-related deaths. Synthetic Lethality (SL) offers a potential solution to this problem. SL involves the simultaneous suppression of two genes, leading to the death of cancer cells while sparing normal cells. By identifying and targeting specific vulnerabilities in cancer cells, SL can induce lethal events, potentially overcoming drug resistance and improving the efficacy of existing treatments. To this end, we retrieved a pool of deleted oncogenes in metastases from a pan-cancer metastatic solid tumor study contained in cBioPortal [39]. This comprehensive study encompasses whole-exome and -transcriptome sequencing of 500 adult patients with metastatic solid tumors and primary normal pairs of diverse lineage and biopsy sites [40]. The gene selection was based on two specific criteria: an expressive state indicating deletion in the metastatic phenotype (i.e., “Variant Type” = “DEL”) and an oncogenic classification (i.e., “Is Cancer Gene (OncoKB)” = “Yes”), resulting in a refined subset of 472 genes. These genes were subsequently intersected with SynLetDB [41], extrapolating couples in which only one of the two genes was part of the previously selected pool. The list of counterpart genes, 2118 in total, was the starting point around which we built the network for the prediction of new drug–target associations. Then, as described in Section 4.3.4, we focused on a subset of 348 genes in SL with genes that, when mutated, may cause metastases. From this subset, we identified 853 FDA-approved compounds cataloged on DrugBank [42] targeting at least one of the 348 genes, resulting in 1662 gene–drug connections. We then applied traditional NMTF and our two-tower architecture approach to predict novel associations. NMTF played a crucial role by enabling the decomposition of complex data matrices into lower-dimensional representations. This factorization method helped in capturing the latent features of the data, which are essential for understanding the intricate relationships between genes and drugs. NMTF provided a foundational baseline for our analysis, allowing us to benchmark the performance of our two-tower architecture against a traditional method. By revealing the inherent structure of the data, NMTF facilitated the identification of potential new drug–target interactions, which were subsequently validated and refined using more advanced techniques. Table 1 demonstrates the superior performance of the two-tower architecture compared to the NMTF-based method. Firstly, it is evident that an embedding length of 25 generally yields better results than a length of 10 (AUROC improves by 4.5% on average). On the same task, the best NMTF performance is obtained when only drug side data are considered. Overall, the best results are obtained with an embedding size of 25 when employing XGBoost as the predictive model, decisively surpassing all NMTF configurations in performance (Figure 1).

We also validated our approach using the LOO technique to simulate the potential for discovering drugs for genes without known associated drugs (LOO Genes) and new targets for novel drugs (LOO Drugs). Again, embeddings of a larger size tend to perform slightly better, with an average improvement of 4.4% in terms of AUROC. Furthermore, in the case of LOO validation, the two-tower architecture performs decisively better than NMTF, with an average improvement of 15.6%.

Additionally, to ensure a comprehensive evaluation of our model’s performance, we compared it not only against the traditional NMTF approach but also using alternative embedding generation methods. This comparison aimed to validate the robustness and efficacy of our two-tower architecture across different data representation techniques. We first employed Singular Value Decomposition (SVD), another matrix factorization technique, to generate embeddings. SVD is a well-established method for dimensionality reduction and data compression, which decomposes a matrix into singular vectors and values, capturing the most significant features of the data. When the embeddings thusly generated were concatenated and input into a finely tuned RF, LR, and XGBoost, we achieved an AUROC of 0.93, 0.93, and 0.74, respectively. These results are comparable to those achieved using NMTF-based factorization, indicating that SVD can serve as a viable alternative for embedding generation in this context. Moreover, we explored the use of a neural autoencoder (AE) to generate embeddings. Autoencoders are a type of neural network designed to learn efficient codings of input data by training the network to output a reconstruction of the input. The encoded, lower-dimensional representation captures essential features of the input data. When these autoencoder-generated embeddings were concatenated and input into a finely tuned XGBoost model, we achieved an AUROC of 0.81. This result suggests that, while the autoencoder-based approach was less effective than SVD and traditional NMTF, it still offers a valuable alternative method for embedding generation, particularly in capturing non-linear relationships in the data.

Several noteworthy observations emerge. Firstly, the two-tower approach, which can integrate embeddings from different generation methods, clearly outperforms the traditional NMTF-based predictor, aligning with expectations. Furthermore, the superiority of non-linear algorithms over linear ones is evident, as complex relationships and patterns are better captured by more powerful models. Lastly, our hypothesis is validated: when adding too many layers to the graphs, the performance of NMTF evaluated on a single-target matrix tends to decrease. This phenomenon is mitigated by the two-tower approach.

Overall, the robustness of our two-tower architecture across various embedding generation methods underscores its potential for wide applicability in biomedical data analysis.

**Literature validation:** To validate the biological significance of our study and validate the reliability of our predictions, we conducted a comprehensive literature review of our new drug–target predictions. This step is crucial to ensure that our computational models align with existing biological knowledge and can potentially guide future experimental research. The 10 highest scoring predictions are detailed in Table 2. Interestingly, among the new predictions, there are several antidepressants. Zheng et al. [43] reviewed the carcinostatic effect of 10 different antidepressants, which have been showed to be able to influence the metabolism of tumor cells through mechanisms such as cell apoptosis, antiproliferative effects, mitochondria-mediated oxidative stress, DNA damaging, changing of immune response and inflammatory conditions, and inhibition of multidrug resistance. Among these drugs, we can find Mirtazapine, which exhibits anti-tumor properties through various mechanisms, such as the inhibition of tumor growth by activating the immune response, increasing serotonin concentration, and elevating IL-12 levels [44]. Additionally, this drug, by enhancing Lin-7 C expression, promoting TNF-alpha levels, and activating the Lin-7 C/β-catenin pathway, proved to be effective against metastases and to have antimetastatic activity [45,46]. It further induces cancer cell death through a Ca^2+^-dependent mechanism and inhibits tumor growth by acting as a histamine receptor 1 antagonist, disrupting the histamine-promoted Ca^2+^-mediated activation of the K^+^ channel, and reducing CXCR4 signaling, which is crucial for tumor cell proliferation and metastasis [47]. Indeed, the targeted DRD2 gene, a dopamine receptor, shows increased expression in many tumors [48] and its inhibition demonstrated effectiveness in suppressing the proliferation of endometrial cancer cells, as well as reducing tumor growth and metastatic potential [49]. Medroxyprogesterone acetate (MPA), a progestin normally used as a contraceptive, has been demonstrated to be effective in the treatment of recurrent and metastatic endometrial carcinoma [50], particularly if coupled with tamoxifen in a phase II study [51]. This drug has been associated with the NR3C1 gene as target, a glucorticoid receptor. Genetic changes of this gene have been associated in several studies to the progress of various solid cancers, including pituitary adenomas [52], colorectal carcinomas [53], gastric cancer [54], and breast cancer [55]. Moreover, high levels of NR3C1 mRNA expression have been correlated with aggressive clinical features and poor prognosis in ovarian cancer [56]. Olsalazine, commonly employed as an anti-inflammatory agent, has demonstrated its capability to impede the progression of colorectal cancer in patients [57,58], and it has been suggested as a versatile anticancer agent with a broad spectrum of activity [59]. It has been predicted to be associated with prostaglandin-endoperoxide synthase 1 (PTSG1), also known as COX1. Kennedy et al. [60] stated that this gene is considerably expressed in colorectal cancer and genes that promote tumor growth and show a strong correlation with COX1 expression are recognized for their role in enhancing mitogenesis, mutagenesis, angiogenesis, cell survival, immunosuppression, and metastasis during the inflammatory processes associated with colorectal cancer. An inhibition of proliferation and migration of colon cancer cells via the suppression of the Wnt signaling pathway by Pizotifen has been shown, making this agent a potential drug for colon cancer treatment. In addition, the repurposing of the antipsychotic Chlorpromazine demonstrated an inhibition of colorectal cancer and pulmonary metastases by an induction of G2/M cell cycle arrest, apoptosis, and autophagy [61].

#### 2.1.2. Parkinson’s Disease Dataset

Delving into the Parkinson’s disease dataset, our focus centered on repurposing drugs for innovative Parkinson’s treatment strategies. This focus stemmed from the critical need within the medical community to find novel solutions to alleviate the complex symptoms and progression of Parkinson’s disease. Characterized by motor dysfunction and other neurological impairments, PD poses significant challenges due to the limited efficacy of current treatments in halting disease progression [62,63]. To ensure that our study was both realistic and reliable, we considered real gene expression data. Specifically, we focused on RNA sequencing (RNA-seq) datasets from patients diagnosed with Parkinson’s disease and matched controls. This approach not only provided insights into the altered molecular pathways in PD but also supported the development of more personalized medicine. By considering the gene expression profiles of individual patients, we aimed to tailor treatments to specific molecular characteristics, thereby potentially improving therapeutic outcomes. Therefore, we accessed publicly available RNA-Seq data sourced from the Gene Expression Omnibus (GEO) repository [64], specifically gathered from post-mortem human brain tissues of both PD patients and healthy control individuals. Specifically, we retrieved five available RNA-Seq datasets (i.e., GSE106608, GSE136666, GSE133101, GSE135036, and GSE134390) with a total of 146 samples from various brain regions (i.e., prefrontal cortex, substantia nigra, putamen, cingulate gyrus, subthalamic nucleus, and amygdala). Following the application of the standardized data processing pipeline, we conducted a differential expression analysis between the PD and healthy control groups, resulting in the identification of 3670 genes with an adjusted *p*-value < 0.05. Then, to make the comparison with NMTF fair, as described in Section 4.3.4, we focused on a subset of 412 genes with statistically significant alterations. Concurrently, we identified 701 approved drugs targeting at least one of those genes.

The results of the computational validation are reported in Table 1. The two-tower architecture clearly outperforms NMTF, exhibiting a performance increase of 20.1% and 14.1% in cross-validation and LOO experiments, respectively. Larger embeddings perform slightly better with an average improvement of 1.3% in terms of AUROC (Figure 2). As in the previous case, we conducted a cross-validation using our two-tower architecture model with SVD for embedding generations. The SVD results yielded performance metrics of 0.93 for RF, 0.93 for XGBoost, and 0.75 for LR. Additionally, we applied a conventional two-tower architecture, where embeddings were learned by a neural autoencoder (AE) and predictions were made by a finely tuned XGBoost classifier, achieving an AUROC of 0.821. The conclusions made on the metastases dataset are corroborated by the findings on this dataset as well.

**Literature validation:** The 10 highest scoring prediction are reported in Table 3. Promethazine has been recognized as a compound with potential neuroprotective properties in the National Institute of Neurological Disorders and Stroke (NINDS) screening program and as having possible neuroprotective effects in a mitochondrial toxin model of PD [65]. Trifluoperazine, a powerful inhibitor of Ca^2+^-transporter proteins, has been shown to selectively reduce one particular α-synuclein species and rescue cells in a model designed to study α-synuclein-mediated toxicity, where human postmitotic dopaminergic neurons were led to gradual death by overexpressing wild-type α-synuclein [66]. Trifluoperazine has also been investigated [67] as a direct antioxidant protector for brain plasma membrane Ca^2+^-ATPase under oxidative stress conditions. Both of these drugs have been linked to the target HTR2A, the 5-hydroxytryptamine receptor 2A, which belongs to the receptor family for serotonin. Genetic variation in the HTR2A can contribute to the vulnerability to impulse control and repetitive behaviors in PD [68]. Furthermore, a remarkable study suggested that Guanabenz, an alpha-2 adrenergic agonist used to treat hypertension, might be employed as therapeutic agents to elevate parkin levels, whose reduced function appears to be a central pathogenic event in PD, consequently decelerating neurodegeneration in PD and other neurodegenerative conditions [69]. ADRA1D (α-1D adrenergic receptor), the target gene with which Guanabenz has been associated, aligns with recent discoveries showing the downregulation of the alpha-D1 adrenergic receptor encoded by ADRA1D in the hippocampus of patients with Alzheimer’s disease and dementia linked to Lewy bodies. Additionally, the presence of noradrenergic impairment in PD patients further emphasizes the pivotal role of neurotransmitters beyond dopamine in the pathology of PD [70]. Finally, Topiramate, an anticonvulsant, could be taken into consideration for a combined therapy with Levodopa since it demonstrated the ability to reduce levodopa-induced dyskinesia, without affecting the antiparkinsonian action of Levodopa [71]. This drug has been coupled with the gamma-aminobutyric acid type A receptor subunit beta1 (GABRB1), which is expressed widely throughout the brain and is implicated in various neurobehavioral diseases. Gamma-hydroxybutyric acid, acting as an agonist, stimulates GABA-B receptors to induce sedative effects [72].

### 2.2. Drug Synergism

Here, we delve into the outcomes yielded by our methodology within the realm of precision medicine. Our primary objective is to evaluate its capacity to forecast the effectiveness of drug combinations through an investigation into drug synergies within cell lines. These cell lines serve as models that mirror patient profiles, utilizing specific gene expression data. This pioneering approach bears the potential to transform healthcare fundamentally by offering profound insights into the interplay between different medications. Consequently, it enables the identification of optimal treatment regimens tailored to individual patient characteristics.

Drug synergism occurs when the combined effect of two or more drugs is greater than the sum of their individual effects [73]. Understanding this phenomenon is crucial for enhancing therapeutic outcomes, reducing dosages, and minimizing adverse effects. In this context, we used a combination of data from different sources to explore drug synergism in cell lines. Our dataset contains values for 101,169 drugs–cell line triplets expressed in terms of ZIP score, involving 1142 distinct compounds and 153 cell lines. The ZIP score is a real number centered around zero; positive values indicate synergism while negative values indicate antagonism. We built the hypergraph in Figure 3c and used NMTF to compute embeddings of dimension *k* = 25, for both cell lines and drugs. One embedding of a cell line and two embeddings of two drugs were concatenated and passed to a regressor/classifier in a multi-tower architecture. We conducted two experiments: (a) a regression analysis to predict the actual value of ZIP for an unseen triplet, and (b) after having dichotomized the ZIP values, we ran a binary classification task.

#### 2.2.1. Regression Analysis of Synergy Scores

We performed the regression analysis on our dataset to predict unseen drug synergy values, employing three different regressors: LR, RF regressor, and XGBoost regressor. To ensure a robust evaluation, we employed five-fold CV. The performance of each model was assessed using Pearson’s and Spearman’s correlation coefficients between predicted values and real ones. LR showed weaker correlations (Pearson’s: 0.273; Spearman’s: 0.212) compared to RF (Pearson’s: 0.744; Spearman’s: 0.657) and XGBoost (Pearson’s: 0.773; Spearman’s: 0.604) regressors. Overall, RF and XGBoost regressors demonstrated superior predictive abilities for drug synergy values, again demonstrating the benefits of adding a non-linear layer. Comparing these findings with our previous work on drug synergism [35], focusing on Pearson’s correlation coefficients, reveals significant insights. In the earlier study, solely utilizing an NMTF-based approach, we achieved Pearson correlation coefficients of 0.757 and 0.760. However, in our current study employing diverse regression models, both random forest and XGBoost regressors demonstrated stronger Pearson correlation coefficients. This suggests that, despite differing methodologies, our current regression models, particularly random forest and XGBoost regressors, achieve comparable or even enhanced performance in predicting drug synergy values based on Pearson correlation coefficients.

#### 2.2.2. Prediction of Drug Synergism

To predict synergistic pairs, we dichotomized the ZIP score, assigning a value of 1 to pairs with positive scores and 0 to those with non-positive scores. We conducted a five-fold CV with hyperparameter tuning, evaluating AUROC metrics. LR achieved an AUROC 0.65, while RF and XGBoost attained 0.83 and 0.82, respectively, indicating strong predictive capability (Figure 4). Finally, we employed LOO validation, where, at each iteration, one cell line was excluded from the training, yielding AUROC scores of 0.61 for LR, 0.68 for RF, and 0.69 for XGBoost (Figure 5). Although slightly lower, these results reflect the model robustness even when faced with cell lines that were not considered during training and the semantic value of NMTF-computed embeddings.

#### 2.2.3. Comparison with NMTF

We propose a qualitative comparison with NMTF, considering that drug synergism necessitates a tensor rather than a matrix. In our previous work from 2021, aimed at predicting drug synergism using NMTF [35], we constructed an association matrix with cell lines as rows and drug pairs as columns. Due to the sparse and unbalanced nature of that matrix, we had to limit the search to 15 cell lines (compared to 153 here) and a maximum of 6796 drug pairs (compared to over 1 million here). These considerations demonstrate that, while maintaining high, if not better, performance, our approach is significantly more flexible and can naturally be applied to a wider range of use cases.

## 3. Discussion

As biomedical knowledge advances, the availability of larger and more reliable datasets of heterogeneous omics data for investigation continues to grow. This abundance of data presents an opportunity for numerous data-driven computational analyses aimed at generating novel hypotheses. In this context, tools capable of seamlessly integrating and fusing heterogeneous information to produce semantic representations of entities of interest, upon which AI methods can be applied, hold particular value. In this study, we proposed NMTF as an effective tool for generating semantically meaningful representations of entities by integrating data from diverse sources. Furthermore, we introduced an architecture to leverage these representations for prediction and inference. Computational pharmacology stands out among the disciplines expected to benefit the most from AI in the near future. In this context, we tested our approach across three use cases, two involving classification tasks and one focused on regression. The results demonstrate a strong performance across both scenarios, surpassing that of conventional methods. A comprehensive literature review revealed that the majority of real predictions (i.e., those not present in the training dataset) were corroborated by biological studies, affirming their validity. Moreover, the core of our method, which involves producing dense semantic representations of entities, leads to exceptional flexibility, enabling adoption across a wide range of scenarios.

Recent pharmacological advancements have seen AI methods play a pivotal role, notably in drug repurposing and personalized medicine, offering innovative and cost-effective alternatives to traditional drug development paradigms [5]. Integrating vast heterogeneous datasets, AI techniques such as NMTF have shown promise in predicting novel therapeutic indications and drug–target relationships. However, NMTF’s linear nature and limitations in capturing complex patterns have prompted the development of more robust frameworks. In this work, we present a novel approach, which leverages NMTF within a multi-tower architecture to predict novel therapeutic indications, drug–target relationships, and drug synergism. Through extensive testing, including LOO validation, we demonstrate the effectiveness and versatility of our method across diverse datasets, paving the way for enhanced AI-driven drug discovery and personalized treatment strategies.

The results obtained through our methodology in the field of drug repurposing demonstrate the robustness of our approach. Utilizing semantic embeddings generated by a two-tower architecture employing NMTF, we conducted rigorous evaluations encompassing qualitative and quantitative analyses within different medical contexts. Specifically, we examined data for both cancer and Parkinson’s disease to find better treatments. For cancer, we focused on Synthetic Lethality to find repurposed drugs, highlighting the importance of identifying effective therapies against metastasis. Using gene deletion data from cBioPortal and SynLetDB, we observed that embeddings of larger sizes (*k* = 25) showed better predictive performance than smaller sizes (*k* = 10). For Parkinson’s disease, we analyzed gene expression data from post-mortem brain tissues, highlighting genes with significant alterations and conducting differential expression analysis. Our results confirm what was observed in the cancer dataset, highlighting the effectiveness of larger embeddings (*k* = 25) in predictive accuracy. In both cases, our approach outperformed the traditional NMTF approach in predicting drug–target associations.

These analyses not only confirmed the effectiveness of our methodology but also unveiled a plethora of potential new drug–target pairs for further exploration and experimental validation. The consistently superior performance of our approach marks a promising advancement in predicting novel associations crucial for advancing treatment strategies in complex diseases like cancer metastases and PD. It also lays the groundwork for application in the field of precision medicine.

Our study also explored drug synergism prediction. Regression analysis revealed varying performance in predicting drug synergy values, with random forest and XGBoost regressors exhibiting superior predictive abilities compared to linear regression. Additionally, our classification task demonstrated the models’ high capability in distinguishing synergistic drug pairs from non-synergistic ones, with random forest and XGBoost achieving particularly intriguing AUROC values of 0.83 and 0.82, respectively.

To ensure a comprehensive evaluation, we compared our two-tower architecture with alternative embedding generation methods, including SVD and neural AE. While SVD, a classical method for dimensionality reduction, provided results comparable to NMTF, AE, designed to capture non-linear relationships, showed slightly lower performance. These findings underscore the versatility of our approach in integrating various embedding techniques.

The comprehensive literature validation of our drug–target predictions, both in cancer metastases and PD, underscores the potential clinical significance of our computational models. For instance, Mirtazapine and Medroxyprogesterone acetate, identified as potential treatments for cancer metastases and recurrent endometrial carcinoma, respectively, suggest that repurposing existing drugs could provide new therapeutic options in oncology [44,50]. Additionally, our predictions point to promising strategies such as targeting DRD2 for tumor suppression and validating Olsalazine and Pizotifen for colorectal cancer through their effects on Wnt signaling and COX1 [57,61]. These results illustrate the model’s capacity to identify novel drug targets and mechanisms beneficial for precision medicine. In Parkinson’s disease, predicted neuroprotective agents like Promethazine and Trifluoperazine, linked to serotonin receptor HTR2A, could lead to new treatments that slow disease progression [65,66]. Furthermore, Guanabenz shows potential in elevating parkin levels to combat neurodegeneration [69], and Topiramate paired with GABRB1 may help manage levodopa-induced dyskinesia [71].

These findings have significant practical implications. First, they highlight the potential for repurposing existing drugs, which can drastically reduce the time and cost associated with drug development compared to de novo drug discovery. This approach can be particularly valuable in treating complex diseases like cancer and PD, where treatment options are limited and often come with significant side effects. Second, the validated drug–target pairs suggest a pathway towards more personalized medicine. By tailoring therapies based on specific genetic and molecular profiles—such as targeting DRD2 in tumors with high dopamine receptor expression or employing drugs that modulate specific neurotransmitter systems in PD—treatment strategies can be more precisely aligned with individual patient profiles. This personalization could improve efficacy and reduce adverse effects, a critical consideration in both oncology and neurodegenerative disease treatment. Lastly, the robust performance of our two-tower architecture across different datasets and conditions underscores its potential as a versatile tool in biomedical research. The model’s ability to integrate various data types and generate meaningful predictions indicates its broader applicability in other diseases and conditions beyond cancer and PD, simply by adding or replacing datasets within the network that contain information relevant to the area of interest for the specific study. This flexibility is crucial for advancing AI-driven drug discovery and could facilitate the development of new therapeutic strategies that are more effective and better tailored to individual patient needs. In conclusion, our study not only provides a solid foundation for future experimental validation and clinical trials but also exemplifies the transformative potential of computational approaches in modern medicine. As we continue to refine and expand our models, they will likely play an increasingly critical role in uncovering novel therapeutic targets and optimizing treatment regimens, ultimately contributing to the advancement of personalized and precision medicine.

## 4. Materials and Methods

### 4.1. Data

Here, we provide a thorough explanation of the data model, elucidating its structure and outlining the relationships between different elements. Simultaneously, we report the data sources contributing to our use cases.

#### 4.1.1. Semantic Multipartite Graph

A graph G=〈N,E〉, where *N* denotes a set of nodes and *E* a set of edges between the nodes in *N*, is said to be semantic multipartite if the following conditions are met:The set of nodes *N* can be split into *n* subsets $N_{1},N_{2},\ldots ,N_{n}$, such that $N_{i}\cap N_{j}=\emptyset$ if i≠j and $\overset{n}{\underset{i=1}{⋃}}N_{i}=N$; furthermore, each subset $N_{i}$ is linked to a semantic category (e.g., genes or pathways);It is possible to assign a semantic meaning to connections between elements of two sets (e.g., the gene represented by $n_{x}$ being a target of the drug represented by $n_{y}$).

Given any pair of subset of nodes $N_{i}$ and $N_{j}$, the matrix $R_{ij}\in {\left\{0,1\right\}}^{|N_{i}\left|\times \right|N_{j}|}$ is called association matrix if, for each pair of nodes, $\langle i_{x},j_{y}\rangle \in N_{i}\times N_{j}$, $R_{ij}\left[x,y\right]=1$ if $i_{x}$ is linked $j_{y}$, 0 otherwise. A semantic multipartite graph can be unequivocally represented by the collection of its non-null association matrices. We use the graph in Figure 3a to predict novel drug–gene association in two different uses cases. Its nodes are divided in seven semantic subsets, namely Genes, Drugs, GO Terms, Pathways, ATC, Categories, and Classes. Edges can be linked to semantic relationships; e.g., edges between Genes and GO Terms indicate that the associated genes are annotated to the corresponding ontological terms. Figure 3c illustrates the graph utilized in the third use case, focusing on identifying pairs of synergic antitumor drugs for a specified cell line. It is noteworthy that the Cell Line and Drugs form a hypergraph, with each cell line being associated with two drugs. Consequently, this part of the graph cannot be adequately represented by a matrix and, instead, requires a tensor.

#### 4.1.2. Data Sources

The computational architecture underpinning this research is predicated on several core entities, interconnected among them via distinct categories of relationships. This section delineates all the databases and the encompassed information that have been employed. The information regarding the *Drugs* in Figure 3a are imported from DrugBank [42], a comprehensive and extensively curated database that serves as a valuable resource in the field of pharmacology and drug research, allowing to search among more than 500,000 drugs and drug products. It offers exhaustive information on 16,558 drugs, including their chemical structures, pharmacology, mechanisms of action, and clinical details. Specifically, we retrieved FDA-approved drugs, along with the associated fourth tier of the Anatomical Therapeutic Chemical (ATC) classification system, which categorizes drugs across five levels based on their effects on specific organs or systems and their therapeutic, pharmacological, and chemical properties, their Categories, and their Classes. The construction of the three association matrices $R_{DA}$, $R_{DT}$, and $R_{DC}$ can be done straightforwardly.

The set of Genes of the graph in Figure 3a are linked to the set of GO Terms and a set of curated Pathways, comprising pathways from BioCarta [74], KEGG [75], PID [76], REACTOME [77], and WikiPathways [78]. Both the set of GO Terms and Pathways have been retrieved from the Molecular Signature Database (MSigDB) [79], a repository containing tens of thousands of annotated gene sets, categorized into collections for human and mouse studies. Each gene set represents a set of genes grouped together as belonging to the same mechanism.

Concerning Figure 3c, the cell lines are associated with the RPKM scores of each gene obtained through RNA-Seq analysis and retrieved by the Cancer Cell Line Encyclopedia (CCLE) [80]. This weighted bipartite graph is represented as the matrix $R_{GC}$ with values in $R_{\geq 0}$. Harmonized synergy value of pairs of drugs for different cell lines from DrugCombDB [81] are expressed in terms of Zero Interaction Potency score (ZIP) [82]; thus, each hyperedge has a value in R.

### 4.2. Prediction of Unknown Links

Here, we focus on methods to predict new edges and the existence or the value of hyperedges.

We outline a method called Non-Negative Matrix Tri-Factorization (NMTF), initially for semantic bipartite graphs and then extended to the general case. However, NMTF is not able to perform predictions on hypergraphs. Next, we introduce our proposed approach, which utilizes NMTF to generate dense semantic representations for two sets of elements. These embeddings, along with known links between entities, are used to train classifiers for subsequent predictions. We demonstrate that the extension to hypergraphs is straightforward.

### 4.3. Methods

#### 4.3.1. NMTF for Semantic Bipartite Graphs

We consider a semantic bipartite graph such as the one connecting the nodes within Genes and *Drugs*. This graph is unequivocally represented by the association matrix $R_{GD}$. The NMTF method can be applied to $R_{GD}$ in order to infer novel links between *Genes* and *Drugs*. It decomposes the association matrix in three lower rank positive matrices $U\in R_{\geq 0}^{\left|\mathrm{Genes}\right|\times k_{g}}$, $S\in R_{\geq 0}^{k_{g}\times k_{d}}$, and $V\in R_{\geq 0}^{\left|\mathrm{Drugs}\right|\times k_{d}}$, with $k_{g}<\mathrm{Genes}$ and $k_{d}<\mathrm{Drugs}$. The decomposition is to minimize the Frobenius norm of the difference between $R_{GD}$ and the product of the three factorization matrices
$$L\left(R_{GD}|k_{r},k_{l}\right)={\left|R_{GD}-USV^{⊤}\right|}_{Fro}^{2}.$$

The *U*, *S*, and *V* matrices are computed starting from a random initialization and iteratively applying a set of update rules until a stop criterion is satisfied. The resulting factorized matrices capture latent features and relationships within the data. By computing an approximation of the original matrix as ${\tilde{R}}_{GD}=USV^{⊤}$ missing values can be predicted: this process is known as *matrix completion* via matrix factorization.

It is worthy to highlight a significant limitation of NMTF for the inference of new edges. When a node has no connection with other nodes (as exemplified by the two nodes with a thick red border in Figure 3a), the corresponding row/column in the association matrix contains only null values. Consequently, it is not possible to capture latent features, relationships, and patterns associated with them, and, therefore, to predict novel edges for such nodes.

#### 4.3.2. NMTF for Semantic Multiparite Graphs

The extension of NMTF to semantic multipartite graphs is straightforward, as we can interpret a multipartite graph as a composition of bipartite graphs. Therefore, a semantic multipartite graph can be represented by the set R of all the association matrices of its sub-bipartite graphs. We can leverage of NMTF to infer a set of factorization matrices that minimizes the loss function:(1)$$L\left(R,\Theta \right)=\sum_{R_{i}\in R}{\left|R_{i}-U_{i}S_{i}V_{i}^{⊤}\right|}_{Fro}^{2}.$$

It is evident that such a formulation would lead to a collection of independently factorized matrices, where data integration would be absent. To comprehensively utilize all the information in the semantic multipartite graph, we must introduce an additional constraint. This applies whenever three sets of nodes *X*, *Y*, and *Z* are chained by two bipartite graphs. We may distinguish between 3 different cases:If *X* is connected to *Y*, which is connected to *Z*, and if <$U_{X},S_{XY},V_{Y}$> and <$U_{Y},S_{YZ},V_{Z}$> are the factorization of their association matrices $R_{XY}$ and $R_{YZ}$, we need to impose that **$V_{Y}=U_{Y}$**;If *X* is connected to both *Y* and *Z*, and if <$U_{X},S_{XY},V_{Y}$> and <$U_{X}^{\prime },S_{XZ},V_{Z}$> are the factorization of their association matrices $R_{XY}$ and $R_{XZ}$, we need to impose that **$U_{X}=U_{X}^{\prime }$**;If both *Y* and *Z* are connected to *X*, and if <$U_{Y},S_{YX},V_{X}$> and <$U_{Z},S_{ZX},V_{X}^{\prime }$> are the factorization of their association matrices of their association matrices $R_{YX}$ and $R_{ZX}$, we need to impose that **$V_{X}=V_{X}^{\prime }$**.

In this framework, it is possible to derive the set of updated rules that, starting from a random initialization of the factorization matrices, computes their values to minimize the objective Function (Equation 1). For each association matrix $R_{XY}$, the update rules for the three decomposition matrices are as follows: $$U_{X}\leftarrow U_{X}⊙\frac{\sum_{Q}R_{XQ}V_{Q}S_{XQ}^{⊤}+\sum_{Q}R_{QX}^{⊤}U_{Q}S_{QX}}{\sum_{Q}U_{X}S_{XQ}V_{Q}^{⊤}V_{Q}S_{XQ}^{⊤}+\sum_{Q}U_{X}S_{QX}^{⊤}U_{Q}^{⊤}U_{Q}S_{QX}}$$
$$V_{Y}\leftarrow V_{Y}⊙\frac{\sum_{Q}R_{QY}^{⊤}U_{Q}S_{QY}+\sum_{Q}R_{YQ}V_{Q}S_{YQ}^{⊤}}{\sum_{Q}V_{Y}S_{QY}^{⊤}U_{Q}^{⊤}U_{Q}S_{QY}+\sum_{Q}V_{Y}S_{YQ}V_{Q}^{⊤}V_{Q}S_{YQ}^{⊤}}$$
$$S_{XY}\leftarrow S_{XY}⊙\frac{U_{X}^{⊤}R_{XY}V_{Y}}{U_{X}^{⊤}U_{X}S_{XY}V_{Y}^{⊤}V_{Y}}$$
where ⊙ ($\frac{\;•\;}{\;•\;}$) denotes the element-wise multiplication (division), and $R_{XQ}$ ($R_{QX}$), $R_{QX}$ ($R_{XQ}$), $R_{YQ}$ ($R_{QY}$), and $R_{QY}$ ($R_{YQ}$) refer to the association matrices of any other set of nodes *Q* associated with either *X* or *Y*.

Inspecting the objective Function (Equation 1), one can identify a further limitation of NMTF applied to semantic multipartite graphs. We consider the situation illustrated in Figure 3a, where the main objective is to enhance the quality and completeness of the target bipartite graph between Genes and Drugs. In the objective Function (Equation 1), the weight of the $R_{GD}$ matrix is proportional to the sum of all of its elements (i.e., the number of edges between Genes and Drugs) and inversely proportional to the sum of the elements of all the other association matrices. This implies that, as we provide additional information to describe the two primary entities, the contribution of the target association matrix to the objective function decreases. Thus, NMTF tends to prioritize fitting the auxiliary matrices, as they overall contain more information and are more relevant for the objective function, scarifying the quality of the reconstruction of the target matrix. This is particularly evident when one of these auxiliary matrices is much larger than the others.

Another intrinsic limitation of NMTF lies in its linearity. However, intricate associations, like those between Drugs and target Genes, might be better addressed by means of non-linear techniques, as their complexity may necessitate the utilization of approaches able to capture non-linear patterns and dependencies for more accurate predictions.

This motivates the introduction of an original method to exploit the power of NMTF for data integration and fusion, while mitigating the mentioned limitations.

#### 4.3.3. Semantic Embeddings and *N*-Tower Architecture

To (partially) overcome the limitations of NMTF in its application to semantic multipartite graphs and hypergraphs for the task of inferring novel edges between nodes of two target sets, we propose an original approach inspired by the two-tower architecture, a widely used technique in recommender systems.

Briefly, the architecure of the method is depicted in Figure 3b,d. Firstly, the method associates an embedding to all the nodes of the two target sets. An embedding is a dense vectorial representation of the corresponding nodes, which preserves semantic similarities and relationships between nodes. Then, for every combination of nodes from the *N* sets, their respective embeddings are concatenated to create a feature vector, and a label is assigned. This label is set to 1 if the nodes are connected in the bipartite graph and 0 otherwise. Finally the generated dataset is used to train a binary classifier and new edges are predicted.

To assign an embedding to each node, we utilized NMTF. Initially, for each target set, we isolated the portion of the semantic multipartite graph related to the set itself by removing the connections between *N* of them. Then, given a target set *T* and any associated set *A*, we rearranged the graph such that *A* is on the left side of *T*. As we work on the association matrices, for the cases in which in the original graph *A* was on the right side of *T*, this simply requires to compute the transpose of the association matrix itself. After this modification, in all the association matrices involving *T*, its elements corresponds to the column of the matrices. This results in *N* sub-graphs, as shown in the leftmost part of Figure 3b,d. For each target set *T*, we then employed NMTF to perform the decomposition of all of its association matrices to set $A_{i}$. It is noteworthy that all these factorizations share the exact same *V* matrix, as imposed by the constraints applied to NMTF in the context of multipartite graphs. Thus, the *V* matrix fuses all the information related to the nodes in the target set that derives from the edges connecting nodes in the associated sets $A_{i}$. Furthermore, there exists a one-to-one correspondence between the rows of *V* and the nodes of the target set; thus, we assigned a dense vectorial embedding to each node.

The benefits of this approach with respect to NMTF are manifold: firstly, it allows to exploit more powerful algorithms for the prediction; moreover, it allows to predict edges with a negative weight (which is not possible with NMTF due to the non-negativity constraint), and, finally, can be easily expanded to hypergraphs of any cardinality.

#### 4.3.4. Evaluation

We conducted extensive experiments across multiple datasets and three use cases to thoroughly validate our approach and assess its performance. Specifically, we aimed to predict novel drug targets for genes associated with PD and metastases, as well as identify synergistic pairs of anti-tumor drugs for various cancer cell lines. For the drug–target relationship prediction, we employed two validation strategies: standard cross-validation and LOO, simulating scenarios where genes or compounds lack associated connections, and reported the results in terms of Area Under the Receiver Operating Characteristic curve (AUROC). Additionally, we compared our results with those obtained using standard NMTF. To this end, we introduced connections between drugs and genes, forming the comprehensive semantic multipartite graph illustrated in Figure 3a. Starting from this semantic multipartite graph, we pruned it to make it suitable for NMTF. Specifically, we only considered drugs linked to one target gene in the set, resulting in the creation of the $R_{GD}$ matrix. To ensure a fair comparison, the $R_{GD}$ matrix containing drugs and genes was used to filter the embeddings employed in the classification tasks, enabling a direct assessment of performance compared to the traditional NMTF method. Finally, we ranked the prediction by score and we conducted a literature review to evaluate the goodness of the top-scoring ones. Furthermore, we compare our results with conventional two-tower architecture employing neural AE and SVD embeddings.

In the prediction of drug synergism, we adopted a three-tower architecture. Since the values within the hypergraph can have negative weights, implying an antagonist relationship between two drugs, our evaluation strategy encompassed both the correlation between the predicted value and the original value and the model’s performance in identifying synergistic pairs, irrespective of their weight. In this case it is not possible to directly compare with NMTF; thus we present a qualitative comparison with a previous work [35].

In our experiments, we conducted a detailed evaluation of two embedding sizes, specifically 10 and 25, across three machine learning models: logistic regression (LR), random forest (RF), and XGBoost. Each model was fine-tuned using a grid search procedure for each use case. Additionally, we explored larger embedding sizes (limited to XGBoost) across all three scenarios to determine whether further increasing the size would lead to improved results. Results are reported in Figure 6 and indicate that increasing the embedding size beyond 25 yields an improvement of less than 1%.

## 5. Conclusions

In conclusion, our study demonstrated the significant potential of using NMTF-based semantic embeddings for integrating heterogeneous omics data, thereby enhancing predictive capabilities in biomedical research. The successful application of our two-tower architecture across various use cases, including drug repurposing and synergism prediction, highlights its versatility and robustness. Notably, our approach outperforms traditional methods in identifying novel drug–target relationships and therapeutic indications, particularly in complex diseases like cancer and Parkinson’s disease. However, it is important to acknowledge certain limitations of NMTF in computing embeddings; firstly, as it is a non-negative method, it requires the weights of all the connections to be positive and this could be a problem with certain types of data. Moreover, as discussed in [38], in presence of many large layers or unbalanced graphs, where some bi-partite subgraphs are significantly larger than others, proper normalization techniques are needed. Overall, our findings suggest that computational methods like ours can play a crucial role in advancing personalized and precision medicine, paving the way for more effective and tailored treatment strategies.

## Figures and Tables

**Figure 1 ijms-25-09576-f001:**
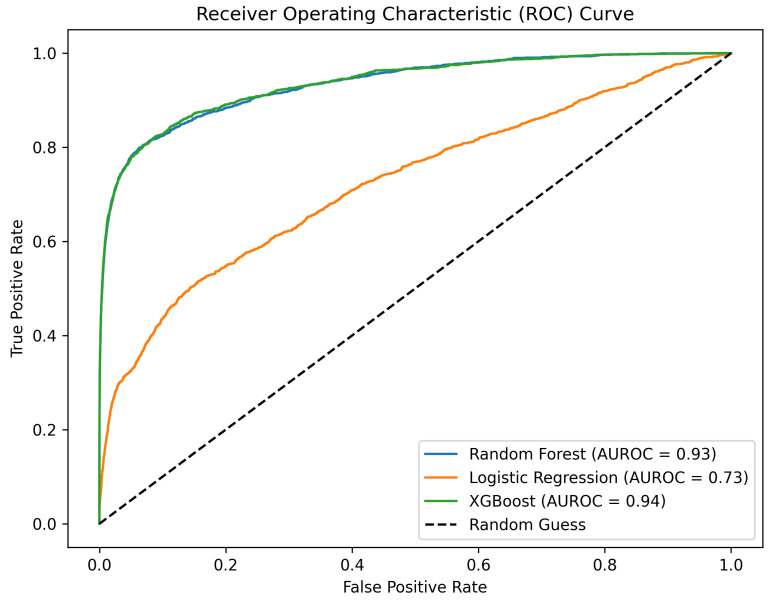
AUROC representation of the implemented method for drug–target prediction with RF, LR, and MLP after hyperparameter tuning for *k* = 25 in metastases dataset.

**Figure 2 ijms-25-09576-f002:**
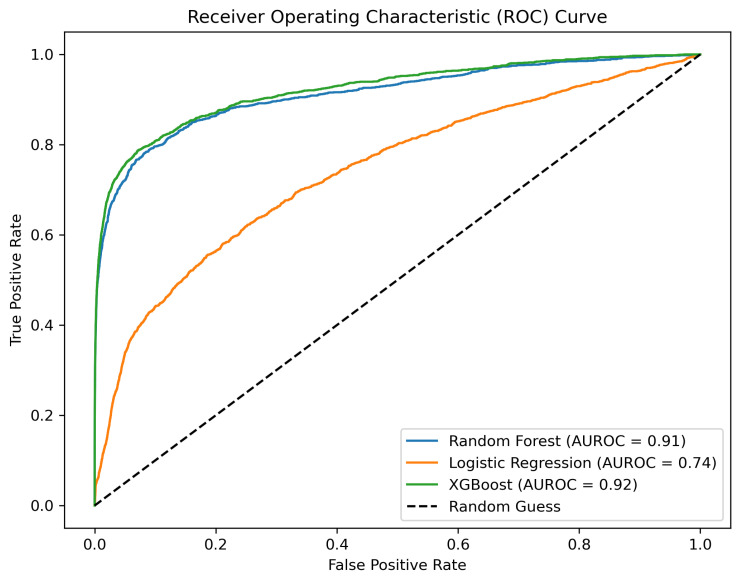
AUROC representation of the implemented method for drug–target prediction with RF, LR, and MLP after hyperparameter tuning for *k* = 25 in Parkinson’s disease dataset.

**Figure 3 ijms-25-09576-f003:**
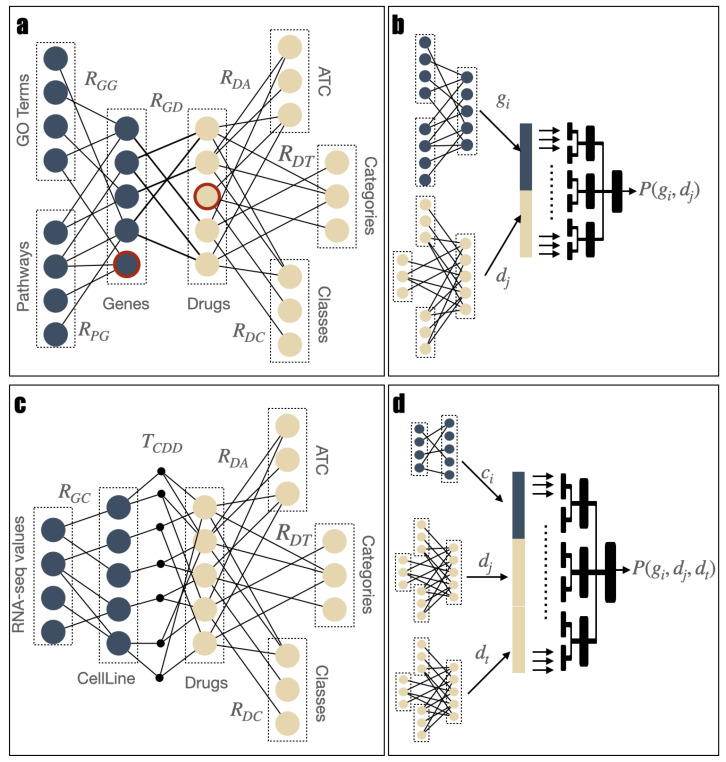
Examples of semantic multipartite graphs and a comparison between using NMTF for prediction versus using NMTF for embedding creation. Panels (**a**,**b**) depict the prediction of novel drug–target relationships. In (**a**), NMTF is directly applied for matrix completion of $R_{GD}$, the association matrix between target genes and drugs. The nodes with a bold red border, represent special instances where the nodes has not previous connection in the drug–target bipartite graph. In (**b**), NMTF is independently applied to target genes and drugs to compute two sets of embeddings, which are then input to a classifier to predict their association. Similarly, panels (**c**,**d**) compare the two uses of NMTF for drug synergism prediction. In this case, the relationships of interest involve three elements (i.e., two drugs and one cell line), making the direct application of NMTF infeasible, as it would require modifying the methods to work with tensors. Conversely, panel (**d**) demonstrates that we can easily adapt the method by inputting the concatenation of three embeddings into the classifier.

**Figure 4 ijms-25-09576-f004:**
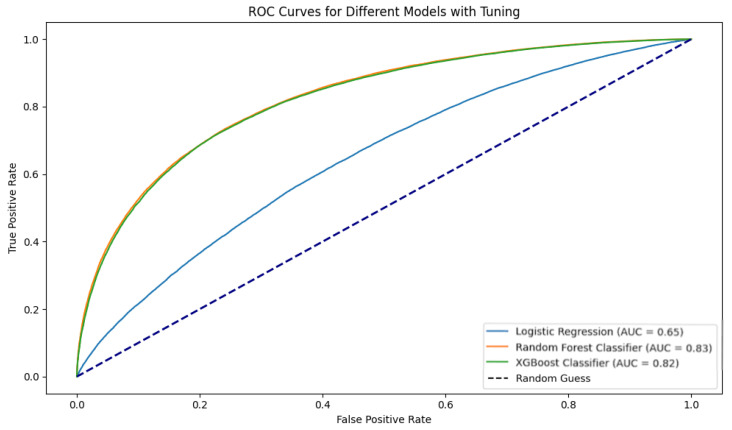
AUROC representation of the implemented method for drug synergism prediction.

**Figure 5 ijms-25-09576-f005:**
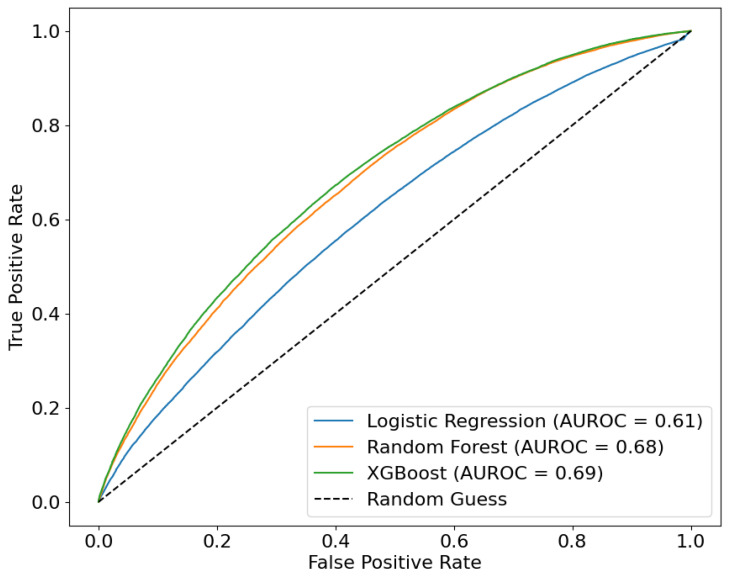
AUROC representation of the implemented method for LOO of cells in drug synergism prediction.

**Figure 6 ijms-25-09576-f006:**
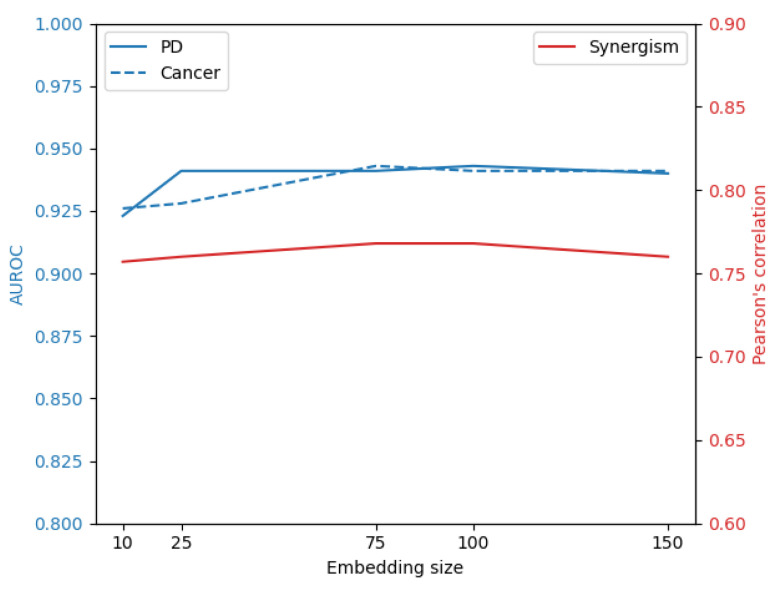
Changes of computational assessed score across the three use cases with XGBoost for the three use cases.

**Table 1 ijms-25-09576-t001:** Results for the metastases and PD datasets are presented. The two-tower architecture was assessed with logistic regression (LR), random forest (RF), and XGBoost (XG) classifiers for both embedding sizes. NMTF was evaluated in four configurations: solely the target association matrix (S), incorporating side data only for genes (G), only for drugs (D), and for both genes and drugs (G+D). Each dataset underwent three experiments: (Full) a 5-fold cross-validation on the target association matrix, (LOO Genes), and (LOO Drugs) a leave-one-out validation on genes and drugs, respectively. Results are reported in terms of AUROC. The best outcome for each dataset and task is highlighted in bold.

Dataset	Task	Two-Tower Architecture						NMTF			
		Embedding Size = 10			Embedding Size = 25						
		LR	RF	XG	LR	RF	XG	S	G	D	G+D
Cancer	Full	0.682	0.904	0.923	0.734	0.937	**0.941**	0.632	0.596	0.796	0.666
	LOO Genes	0.683	0.786	0.767	0.660	**0.828**	0.816	-	0.735	-	0.700
	LOO Drugs	0.664	0.881	0.881	0.707	0.899	**0.904**	-	-	0.755	0.763
PD	Full	0.703	0.923	0.926	0.744	0.912	**0.928**	0.657	0.656	0.773	0.701
	LOO Genes	0.675	0.835	0.828	0.675	**0.858**	0.818	-	0.784	-	0.751
	LOO Drugs	0.684	0.865	0.872	0.715	0.868	**0.881**	-	-	0.716	0.742

**Table 2 ijms-25-09576-t002:** Top 10 predictions on metastases dataset.

Random Forest		XGBoost	
Drug	Pred. Target	Drug	Pred. Target
Mirtazapine	DRD2	Amitriptyline	DRD2
Dosulepin	DRD2	Olsalazine	PTGS1
Doxepin	DRD2	Norgestrel	NR3C1
Trazodone	DRD2	Oxyphenbutazone	PTGS1
Citalopram	DRD2	Trazodone	DRD2
MPA	NR3C1	Glutamic acid	ABCC8
Flupentixol	HRH1	Zonisamide	CACNA1S
Vortioxetine	DRD2	Pizotifen	DRD2
Butriptyline	DRD2	Chlorpromazine	CHRM4
Norgestrel	NR3C1	Dimetindene	CHRM1

**Table 3 ijms-25-09576-t003:** Top 10 predictions on PD dataset.

Random Forest		XGBoost	
Drug	Pred. Target	Drug	Pred. Target
Prochlorperazine	HTR2A	Fostamatinib	TAB1
Promethazine	HTR2A	Guanabenz	ADRA1D
Olanzapine	ADRA2B	Topiramate	GABRB1
Fluoxetine	HTR2A	Amitriptyline	ADRA2B
Trifluoperazine	HTR2A	Zopiclone	GABRB1
Methoxamine	ADRA2B	Butabarbital	GRIN2A
Midodrine	ADRA2B	Methysergide	ADRA2B
Olanzapine	ADRA1D	Rotigotine	HTR2A
Milnacipran	HTR2A	Zopiclone	GABRA4
Rotigotine	HTR2A	Zopiclone	GABRG2

## Data Availability

For the drug information, data were obtained from the DrugBank database and are available with permission from DrugBank. For the Parkinson’s disease dataset, gene expression data were derived from the Gene Expression Omnibus repository, reference numbers GSE106608, GSE136666, GSE133101, GSE135036, and GSE134390. For the metastases dataset, genes were obtained from the cBioPortal database intersected with SynLethDB and are available at https://www.cbioportal.org, (accessed on 18 July 2024) https://synlethdb.sist.shanghaitech.edu.cn (accessed on 18 July 2024). For the drug synergism, cell lines were obtained via the Cancer Cell Line Encyclopedia at https://depmap.org (accessed on 18 July 2024), whereas the synergy value of pairs of drugs for different cell lines was obtained from DrugCombDB at https://drugcombdb.denglab.org (accessed on 18 July 2024).

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
