# Peer review of "Non-Negative Matrix Tri-Factorization for Representation Learning in Multi-Omics Datasets with Applications to Drug Repurposing and Selection"

_ijms, 2024, doi:10.3390/ijms25179576_

Round 1

Reviewer 1 Report

Comments and Suggestions for Authors

In this paper, the authors propose non-negative matrix triple decomposition as a valuable tool for integrating and fusing data and representation learning. It is shown how representations learned through non-negative matrix triple decomposition can be effectively utilized by traditional artificial intelligence methods. The authors conduct extensive experiments to evaluate the performance of the proposed method, which shows a significant improvement in comparison with traditional methods. The new drug target predictions are also validated in the literature, further confirming their effectiveness. However, this article suffers from the following problems. 

1. NMTF is a linear approach and may not be able to capture complex nonlinear associations, especially in complex relationships between drugs and target genes. It is recommended to introduce nonlinear methods such as graph neural networks (10.1109/TFUZZ.2023.3338565) to complement NMTF to capture more complex nonlinear patterns and dependencies.

2. NMTF may not be able to deal well with unconnected nodes whose corresponding rows/columns in the association matrix are all zeros, resulting in an inability to capture potential features and relationships associated with these nodes. It is suggested to introduce other complementary matrix methods to better handle sparse graph structures and capture features of unconnected nodes.

3. The article mainly compares traditional NMTF methods and multi-tower architectures but lacks comparisons with other state-of-the-art methods such as graph neural networks or other deep learning models. This may limit the reader's full understanding of the relative advantages of the proposed methods. More baseline models and comparison experiments are introduced, which can better demonstrate the performance of the proposed method in different scenarios.

4. The article shows the experimental results in detail but lacks an in-depth explanation and discussion of the results, especially the differences and reasons for the results under different experimental settings. For example, although the effect of embedding size on the results is mentioned, there is no in-depth analysis of why larger embedding sizes perform better. It is recommended that a more in-depth explanation and discussion of the experimental results be added.

5. To better demonstrate the validity of the method it is recommended to include a case study using real data to demonstrate the validity of the method for real data.

Comments on the Quality of English Language

Quality of English language is good.

Reviewer 2 Report

Comments and Suggestions for Authors The authors have applied their customized Non-Negative Matrix Tri-Factorization (NMTF) method to AI-based drug repurposing. The NMTF can learn semantic representations of entities within complex multipartite graphs, which enable accurate predictions of novel drug-disease interactions, drug targets, etc. The paper is well-structured. It starts from the review of relevant research works, which leads to problem formulation. The next section presents the proposed approach, which is then experimentally verified. The experiments were aimed at verifying the efficacy of the proposed approach based on NMTF in the areas of drug-target prediction, and drug synergism. Two datasets were used: Metastases Dataset, and Parkinson’s Disease Dataset. Please address the following issues: 1. Please clearly show the paper's main contributions in the Introduction section. 2. Please explain what would have to be changed in the proposed approach to make it applicable in other areas. 3. Please show clearly in the results what is the contribution of NMTF to the quality of the results obtained. What would these results be if this approach was not used? 4. What other methods would produce similar results to your approach? 5. Please analyze the limitations of the proposed method.

Comments on the Quality of English Language

Please improve the English language, especially in terms of grammar and style.

Reviewer 3 Report

Comments and Suggestions for Authors

The manuscript provides a comprehensive and detailed analysis of using Non-Negative Matrix Tri-Factorization (NMTF) for drug repurposing and selection. The methodological rigor and extensive validation through different datasets, including cancer metastasis and Parkinson's disease, highlight the robustness and potential applicability of the proposed approach. However, a few aspects could benefit from further elaboration. Firstly, the manuscript could enhance clarity by providing more detailed explanations of the selection criteria for the genes and drugs included in the study, particularly regarding the datasets for cancer and Parkinson's disease. Additionally, while the two-tower architecture and its comparison with traditional NMTF are well-explained, a more in-depth discussion on the limitations and potential improvements of this architecture would be valuable. Furthermore, the discussion section could be strengthened by including more practical implications of the findings and how they might influence future drug discovery and personalized medicine. Lastly, considering the complexity of the computational methods used, it would be helpful to include more visual aids, such as diagrams or flowcharts, to illustrate the workflow and key processes involved in the study.

Round 2

Reviewer 1 Report

Comments and Suggestions for Authors

After reviewing the authors' responses, I am not satisfied with their approach to revising this manuscript according to the comments. Here are my main concerns:

1. Graph neural networks have been widely applied to drug repurposing. I am unclear why the authors consider the application of GNNs beyond the scope of their work. This omission needs clarification.

2. To demonstrate the superiority of the proposed model, it is highly recommended that the authors compare their model with state-of-the-art (SOTA) models in the field of drug repurposing. However, the authors only included comparisons with two additional algorithms, SVD and VE, which were published several years ago. A more comprehensive comparison with recent and relevant models is necessary.

3. Regarding the impact of embedding length on the performance of the proposed model, the authors should conduct more experiments to determine whether there is an upper limit for the embedding length. It is insufficient to simply conclude that a longer embedding length will improve prediction performance. Detailed experimental results and analysis are required to support such a claim.

Author Response

Reviewer: After reviewing the authors' responses, I am not satisfied with their approach to revising this manuscript according to the comments. Here are my main concerns:

Response: We regret that the Reviewer is not satisfied with the current version of the manuscript. We did our best to improve our work based on the Reviewer's suggestions and have carefully considered each comment to enhance the quality of the manuscript. Below, we explain our approach and reasoning in addressing the Reviewer's concerns. We sincerely hope the Reviewer understands our constraints and the motivation behind our focused approach. We greatly appreciate and thank the Reviwer for their valuable feedback and respectfully request a wise and fair evaluation of our efforts.

Comment 1. Graph neural networks have been widely applied to drug repurposing. I am unclear why the authors consider the application of GNNs beyond the scope of their work. This omission needs clarification.

Response 1. Response: We thank the Reviewer for this observation. While many studies using GNNs focus on modeling molecules (which we did not include as related work, since our focus is on semantic annotations rather than molecular information), there are indeed some works that propose GNNs for link prediction in graphs similar to ours. We have now cited the relevant works in the introduction of the manuscript.

Comment 2. To demonstrate the superiority of the proposed model, it is highly recommended that the authors compare their model with state-of-the-art (SOTA) models in the field of drug repurposing. However, the authors only included comparisons with two additional algorithms, SVD and VE, which were published several years ago. A more comprehensive comparison with recent and relevant models is necessary.

Response 2. We appreciate the Reviewer's suggestion to compare our model with more recent state-of-the-art (SOTA) models in the field of drug repurposing. However, our choice to compare our approach with two well-known algorithms, SVD and VE, was deliberate. By doing so, we aimed to provide readers with a clear and accessible benchmark to understand where our method stands in relation to established approaches.

Furthermore, our focus in this work has been on the practical applicability of our method rather than solely on computational performance. We prioritized literature validation and domain-specific relevance. Also, we belive that, in this domain, evidence-based validation performed by pharmacologists and clinitian experts is much more important than computationally assessed performance. Yet, we argue that demonstrating practical utility and applicability on real-world data often provides more value than purely comparative performance metrics.

Lastly, while we recognize the value of a comprehensive comparison, it would not have been feasible to conduct an in-depth evaluation of multiple SOTA methods within the journal's revision policy, which grants only 10 days for revisions. We hope the Reviewer can understand these constraints and the rationale behind our approach.

Comment 3. Regarding the impact of embedding length on the performance of the proposed model, the authors should conduct more experiments to determine whether there is an upper limit for the embedding length. It is insufficient to simply conclude that a longer embedding length will improve prediction performance. Detailed experimental results and analysis are required to support such a claim.

Response 3. We appreciate this interesting and "on-point" observation. To address this, we run NMTF to produce larger embedding sizes (75, 100 and 150). Results indicate that 25 is nearly optimal and increasing the size of the embeddings does not significantly improve the performance of the framework. On contrast, while computing 25-sized embedding takes approximately 20 minutes (for each entity type), generating embedding of size 150 require more than 12 hours. We added the results in the discussion Section of the manuscript.

Reviewer 2 Report

Comments and Suggestions for Authors

The authors have addressed the most important issues, so the paper can be accepted.

Author Response

Comments 1: The authors have addressed the most important issues, so the paper can be accepted.

Response 1: We thank again the Reviewer for their precious feedback.

Round 3

Reviewer 1 Report

Comments and Suggestions for Authors

All of my concerns have been addressed in this revision.